# Cytokine Profiling of Plasma and Atherosclerotic Plaques in Patients Undergoing Carotid Endarterectomy

**DOI:** 10.3390/ijms25021030

**Published:** 2024-01-14

**Authors:** Daria Potashnikova, Elena Maryukhnich, Daria Vorobyeva, George Rusakovich, Alexey Komissarov, Anna Tvorogova, Vladimir Gontarenko, Elena Vasilieva

**Affiliations:** 1Laboratory of Atherothrombosis, Cardiology Department, A.I. Yevdokimov Moscow State University of Medicine and Dentistry, 127006 Moscow, Russia; 2City Clinical Hospital Named after I.V. Davydovsky, Moscow Department of Healthcare, 109240 Moscow, Russia; georgerusakovich@gmail.com (G.R.);; 3National Research Center “Kurchatov Institute”, 123182 Moscow, Russia; 4Department of Vascular Surgery, National Medical Research Centre of Surgery Named after A.V. Vishnevsky under the RF Public Health Ministry, 117997 Moscow, Russia

**Keywords:** atherosclerosis, atherosclerotic plaque, chronic inflammation, cytokines, chemokines, blood plasma, tissue explants, tissue culture, xMAP, Luminex, qPCR

## Abstract

Atherosclerotic plaques are sites of chronic inflammation with diverse cell contents and complex immune signaling. Plaque progression and destabilization are driven by the infiltration of immune cells and the cytokines that mediate their interactions. Here, we attempted to compare the systemic cytokine profiles in the blood plasma of patients with atherosclerosis and the local cytokine production, using ex vivo plaque explants from the same patients. The developed method of 41-plex xMAP data normalization allowed us to differentiate twenty-two cytokines produced by the plaque that were not readily detectable in free circulation and six cytokines elevated in blood plasma that may have other sources than atherosclerotic plaque. To verify the xMAP data on the putative atherogenesis-driving chemokines MCP-1 (CCL2), MIP-1α (CCL3), MIP-1β (CCL4), RANTES (CCL5), and fractalkine (CX3CL1), qPCR was performed. The *MIP1A* (*CCL3*), *MIP1B* (*CCL4*), *FKN* (*CX3CL1*)*,* and *MCP1* (*CCL2*) genes were expressed at high levels in the plaques, whereas *RANTES* (*CCL5*) was almost absent. The expression patterns of the chemokines were restricted to the plaque cell types: the *MCP1* (*CCL2*) gene was predominantly expressed in endothelial cells and monocytes/macrophages, *MIP1A* (*CCL3*) in monocytes/macrophages, and MIP1B (CCL4) in monocytes/macrophages and T cells. *RANTES* (*CCL5*) was restricted to T cells, while *FKN* (*CX3CL1*) was not differentially expressed. Taken together, our data indicate a plaque-specific cytokine production profile that may be a useful tool in atherosclerosis studies.

## 1. Introduction

Atherosclerosis is a major cause of cardiovascular disease that accounts for the largest number of deaths worldwide [1,2]. Atherosclerotic plaques are formed in the blood vessel wall and represent sites of chronic inflammation, as evidenced by their infiltration by activated macrophages and T cells [3,4], documented antigen-driven B-cell expansion [5], and the presence of autologous antibodies against oxidized low-density lipoproteins [6]. The roles of immune cell populations in atherogenesis have been extensively studied in experimental animal models. Mass cytometry and single-cell RNA sequencing have revealed vast immune cell heterogeneity and, specifically, a large number of macrophage subpopulations in model mouse plaques [7,8]. Similar to humans, the principal role of CD8+ T cells in mouse atherogenesis was demonstrated in CD8-depletion studies [9]. In humans, the late stages of atherosclerosis are associated with the increased accumulation and activation of CD8+ cytotoxic T lymphocytes (CTLs), which comprise the predominant lymphocyte population in advanced plaques [10,11]. These data highlight the importance of aberrant immune activation in atherogenesis and the role of immune mediators as putative targets for atherosclerosis control and prevention.

The activation and targeting of immune cells to different organs are driven by cytokines, a diverse group of signaling proteins [12,13]. In atherogenesis, the major roles in attracting macrophages and T cells to the plaques are attributed to the chemokine receptors CCR2, CCR5, and CX3CR1, and their respective ligands: MCP-1 (CCL2) [14,15,16], MIP-1α (CCL3)/MIP-1β (CCL4)/RANTES (CCL5) [17,18,19], and fractalkine (CX3CL1) [20,21]. However, the differential roles of these chemokines and their possible partners in driving inflammation at atherosclerosis sites remain to be elucidated. Animal models do not always adequately reproduce the composition of human immune infiltrates and the structure of the arterial wall [22,23], whereas in vitro systems of cultured or co-cultured human cells do not cover the entire range of signaling interactions occurring within human atherosclerotic plaques [24,25]. The levels of cytokines in the blood plasma of patients often provide clues to immune signaling aberrations in disease; however, they only reflect major systemic changes in the organism. The moderately expressed cytokines or those predominantly bound to the endothelial surface or extracellular matrix [26,27] may be lost during whole plasma analysis. Hence, there is a need for a more comprehensive atherosclerotic plaque model that would allow local assessment of cytokine production.

We have previously described an ex vivo model of sub-cultivated human atherosclerotic plaque explants, which retained the general plaque cytoarchitecture, viable predominant plaque cell populations, and cytokine production capacities for 19 days of sub-cultivation [28]. The aim of this study was to compare systemic cytokine production (in the blood plasma of patients with atherosclerosis) and local cytokine production (in the culture medium conditioned by plaque explants from the same patients), and to further confirm the expression of the selected cytokines in plaques by qPCR.

To accomplish this, screening was performed using xMAP technology with a commercial 41-plex cytokine kit. To normalize the xMAP data in this study, two reference cytokines were chosen based on their representation stability in both types of biological material. The expression of reference cytokines in the plaque directly after endarterectomy was confirmed using qPCR. The normalized amounts of cytokines allowed us to obtain a comprehensive signaling pattern in the blood plasma and plaques of patients with atherosclerosis. The cytokines selected for in-depth analysis were chemokines commonly associated with attracting macrophages and CTLs to atherosclerotic plaques, namely MCP-1 (CCL2) [14], MIP-1α (CCL3) [17], and MIP-1β (CCL4) [18]. These were upregulated in the conditioned medium compared to the blood plasma, whereas another putative migration-driving chemokine, RANTES (CCL5) [19], was upregulated in the plasma compared to the conditioned medium.

We confirmed the expression patterns of these chemokines in endarterectomized plaques via qPCR as previously described [29]. Moreover, in this study, we were able to attribute their expression to distinct types of producing cells by sorting macrophages, T lymphocytes, and endothelial cells from endarterectomized atherosclerotic plaques for qPCR analysis. The data obtained provide useful clues to the mechanism of chronic inflammation development and immune cell attraction to atherosclerotic lesions.

## 2. Results

### 2.1. Cytokine Levels in Blood Plasma and Conditioned Medium from Atherosclerotic Plaques

The 41 multiplexed cytokines were measured for 23 paired samples of blood plasma and plaque-conditioned medium from patients with atherosclerosis (Appendix A—Table A1). The workflow is illustrated in Figure 1A. Cytokine concentrations were provided in pg/mL based on the standard curves in each plate. Values that fell above or below the detection range of the method were considered initially missing. The distribution of missing data in the blood plasma and the plaque-conditioned medium is presented in Figure 1B and Figure 1C, respectively. 

Only five cytokines out of forty-one fell into the detection range in both the blood plasma and the plaque-conditioned medium samples from all patients: PDGF-AA, MIP-1β (CCL4), MDC (CCL22), IP-10 (CXCL10), and TNF-α. Therefore, they were assessed as putative normalization controls in the experiment. 

### 2.2. The Choice and Verification of the Normalization Control Cytokines

The criteria for normalization controls included the detectable expression in each sample, minimal variance, and no difference in expression between groups after normalization.

The variance of the putative normalization control cytokines PDGF-AA, MIP-1β (CCL4), MDC (CCL22), IP-10 (CXCL10), and TNF-α was assessed using R 4.0.5 with the GeNorm algorithm from the “ctrlGene” package (see corresponding Materials and Methods, Section 4). The cytokine ranking according to their average stability and pairwise variations of the ranked control cytokines are presented in Figure 2 A,B. As the ranking was produced via pairwise comparison, the best-ranked pair of cytokines was identified, neither of which was ranked more stable than the other. In our sample set, these were TNF-α and IP-10. To compare the obtained normalization factors NF2, NF3, NF4, and NF5 (calculated as the geometric means of cytokine concentrations, where n is the number of putative control cytokines, ranked from the most stable to the least stable according to Figure 2A), and the two most stable control cytokines (TNF-α and IP-10), we used Spearman correlation (Figure 2C).

The smallest pairwise variation (Figure 2B), V3/V4 = 0.253, was obtained for the third and fourth cytokines in the ranking. Similarly, NF3 and NF4 showed the highest correlation (R = 0.92) (Figure 2C). This implies that adding a fourth cytokine to the normalization factor does not change it, and NF3 should be used. In our sample set, the three most stably represented cytokines were TNF-α, IP-10, and MDC. The normalization factor NF3 was calculated as the geometric mean of the TNF-α, IP-10, and MDC concentrations in each analyzed sample. However, after normalizing the putative control cytokine levels by NF3, the normalized levels of both TNF-α and MDC appeared significantly different in the plasma vs. culture medium when compared using the Wilcoxon signed-rank test. This contradicted our assumption that the selected control cytokines should not differ after normalization between the two types of biological specimens. 

We compared the levels of the putative control cytokines in the plasma and conditioned medium after normalization by different normalization factors and the single cytokines TNF-α and IP-10, using the Wilcoxon signed-rank test (for the significance, see Appendix A, Table A2). Only TNF-α and IP-10 were not differentially represented in the plasma or the conditioned medium. The other three control cytokines, MDC, PDGF-AA, and MIP-1β (CCL4), were differentially represented regardless of the normalization factor used. Moreover, adding MDC and PDGF-AA to the normalization factor resulted in significant differences in the TNF-α levels between the plasma and the conditioned medium (Wilcoxon test, *p* < 0.05).

The lack of differential representation of TNF-α and IP-10 and the high correlation coefficient between NF2 and NF3 (R = 0.81) led us to omit the third cytokine (MDC) from the normalization factor. Additionally, to prove that the most stable normalization control cytokines were originally present in plaques, we tested the expression of the *MDC*, *TNFA*, and *IP-10* genes in nine plaques immediately after endarterectomy using real-time qPCR. *TNFA* and *IP-10* were expressed in all the specimens, whereas *MDC* was absent in one of the nine (Figure 2D–F). Thus, owing to the differential expression in blood plasma and conditioned medium, and the absence in one of the nine tested atherosclerotic plaque specimens at the mRNA level, MDC was excluded from the normalization factor. NF2, calculated as the geometric mean of the TNF-α and IP-10 concentrations, was used as a normalization factor.

The possible co-regulation of cytokines was assessed using cluster analysis. Clusterization was performed using Ward’s method for cytokines that were suitable for quantitative analysis (Appendix A—Figure A1). The data were clustered without normalization to avoid possible bias; the cytokines in the same cluster were likely to be co-regulated in the experimental system. TNF-α and IP-10 fell within the same cluster together with MIP-1β (CCL4) in the plasma and conditioned medium. In the conditioned medium, RANTES (CCL5) and eotaxin (CCL11) were included in the same cluster. This implies that our two normalization controls may be co-regulated in the plasma and the plaque-conditioned medium.

### 2.3. Comparison of the Normalized Cytokine Levels in Blood Plasma and Conditioned Medium from Atherosclerotic Plaques

The cytokines that were not included in the normalization factor were imputed (see Materials and Methods, Section 4), normalized, and compared between blood plasma and conditioned medium. Cytokines with less than 40% missing and extrapolated values were compared quantitatively using the Wilcoxon signed-rank test. Cytokines with more than 40% missing and the extrapolated values were compared qualitatively using the sign test. 

Of the forty-one cytokines studied, nine were analyzed quantitatively after normalization. Eotaxin (CCL11), RANTES (CCL5), MDC (CCL22), sCD40L, PDGF-AA, and PDGF-AB/BB were differentially elevated in the blood plasma compared to the conditioned medium (Wilcoxon test, p. adj < 0.05). IL-15 and MIP-1β (CCL4) levels were differentially elevated in the conditioned medium compared to the blood plasma (Wilcoxon test, p. adj < 0.05). IL-5 was not differentially represented between the two types of specimens. The results are shown in Figure 3. Importantly, all cytokines that fell in the same cluster as the normalization control cytokines were differentially represented in the two types of specimens.

Among the cytokines analyzed qualitatively using the two-sided sign test, IL-1β, IL-1RA, IL-2, IL-4, IL-6, IL-7, IL-8, IL-10, IL-12 (p40), IL-12 (p70), IL-13, IFN-α2, MCP-1 (CCL2), MCP-3 (CCL7), MIP-1α (CCL3), GRO-α (CXCL1), VEGF, G-CSF, GM-CSF, and TGF-α were elevated in the conditioned medium compared to the blood plasma (Appendix A, Table A3). The data for all cytokines are summarized in Table 1.

### 2.4. Target Chemokine Gene Expression in Atherosclerotic Plaques

Bulk RNA samples from nine atherosclerotic plaques were analyzed for the expression of the major chemokine genes involved in immune cell migration: *MCP1* (*CCL2*), *MIP1A* (*CCL3*), *MIP1B* (*CCL4*), *RANTES* (*CCL5*), and *FKN* (*CX3CL1*). The relative amounts of mRNA were normalized to the *UBC* reference gene, as described previously [29]. The data were compared with nine paired specimens of peripheral blood mononuclear cells (Appendix A, Figure A2). The *MCP1* (*CCL2*), *FKN* (*CX3CL1*), *MIP1A* (*CCL3*), and *MIP1B* (*CCL4*) genes were expressed in the plaque tissue directly after endarterectomy, and *RANTES* (*CCL5*) was expressed at very low levels, barely above the detection limit. The chemokine distribution between the plaque and blood cells was similar to that described previously [29]: *MCP1* (*CCL2*) and *FKN* (*CX3CL1*) were significantly elevated in plaque tissue compared to blood cells, while *RANTES* (*CCL5*) was significantly elevated in blood cells compared to plaque tissue. There was also a trend toward *MIP1A* (*CCL3*) elevation in plaque tissue; however, the differences were not statistically significant, possibly due to the small number of samples. The *MIP1B* (*CCL4*) gene was not differentially expressed in plaque tissue vs. blood cells. To find the cell sources of the selected target chemokines within the atherosclerotic plaques, we further sorted different cell subpopulations for qPCR.

### 2.5. Target Chemokine Gene Expression in Different Cell Subsets of Atherosclerotic Plaques and Peripheral Blood

Populations of plaque endothelial cells, monocytes/macrophages, and T cells, as well as peripheral blood monocytes/macrophages and T cells, were sorted as described in Materials and Methods (the representative sorting plots are provided in Appendix A, Figure A3). The relative normalized amounts of mRNA were compared between different cell subpopulations for each patient (presented as heatmaps in Figure 4). Thus, it was shown that *MCP1* (*CCL2*) was expressed in endothelial cells and monocytes/macrophages of the plaque (nine out of ten cases) and was absent in plaque T cells (eight out of ten cases). In blood, it was expressed in monocytes/macrophages (five out of ten cases) and was absent in T cells in all cases. The *MIP1A* (*CCL3*) gene was expressed in plaque monocytes/macrophages and at lower levels in plaque T cells. The *MIP1B* (*CCL4*) gene was predominantly expressed in plaque T cells and at lower levels in plaque monocytes/macrophages. The *RANTES* (*CCL5*) gene was predominantly expressed in blood T cells; however, it was also present in the population of plaque T cells (seven out of ten cases). *FKN* (*CX3CL1*) was expressed by all the cell subpopulations analyzed. 

For further comparison, the delta-deltaCt values were plotted for each target chemokine in different cell populations (Appendix A, Figure A4). The expression values below the detection limit were replaced by the arbitrary value “0.0001”, which was at least two-fold lower than any delta-deltaCt value obtained for the sample set. The comparison of the delta-deltaCt values allowed us to attribute *MCP1* (*CCL2*) expression equally to plaque endotheliocytes and monocytes/macrophages, and *MIP1B* (*CCL4*) expression equally to plaque monocytes/macrophages and T cells. *MIP1A* (*CCL3*) was predominantly expressed in plaque monocytes/macrophages, while *RANTES* (*CCL5*) was restricted to T cells. *FKN* (*CX3CL1*) had no predominant cell source.

## 3. Discussion

Multiplexed protein screening techniques, among which bead-based immunoassays are very popular [30], have great value for comprehensive immunological profiling and may help uncover the role of immune signaling in atherogenesis [31]. Here, for the first time, we attempted to compare the systemic and local cytokine levels in patients with atherosclerosis by comparing two types of specimens: blood plasma and culture media directly conditioned by plaque tissue explants. To focus on soluble cytokines, we employed the xMAP detection technology, which combines the high sensitivity of ELISA-based methods with high multiplexing potential [32,33]. Soluble proteins can be quantitatively assessed via xMAP over a broad dynamic range. However, approaches to cytokine measurement and comparison in different types of biological material have not yet been developed. Particularly, such novel studies require new data normalization strategies. The total protein content and the fractions of specific proteins differ greatly between the plasma and the conditioned medium; so, normalization by total protein, as is the case in Western blotting and mass spectrometry [34,35], is likely to introduce more bias. Similarly, normalization by cell number is not feasible, as the cell source of plasma cytokines and the cell numbers in explant tissue are difficult to estimate. Therefore, we chose normalization by internal reference: a cytokine or cytokines that showed stable amounts in each sample. To choose the most stable reference cytokines, we confirmed that they fell within the detection limits in each analyzed sample and had the smallest variance in our dataset. Additionally, we chose the reference cytokines that would not be differentially represented in blood plasma vs. conditioned medium, as normalization by such controls could bias the data.

The limiting factor for the reference choice in our case was the plasma levels of cytokines, which demonstrated a low to moderate abundance [36]. Indeed, some proinflammatory cytokines in the assay were not detectable in the absence of acute systemic inflammation, and some chemokine molecules did not enter free circulation, as they mostly bind to the endothelial surface and/or extracellular matrix fibers and drive cell migration in the immobilized state [37]. 

Based on the preliminary data assessment, only five of the forty-one cytokines were present in all the samples of the blood plasma and plaque-conditioned medium: PDGF-AA, MIP-1β (CCL4), TNF-α, IP-10 (CXCL10), and MDC (CCL22). This is not surprising given the biological differences in our specimens, hence the need for multiplex analysis in the primary screening and comparison of such materials. The conditioning of the culture medium by plaque explants was set to 3 days to yield similar amounts of these cytokines. The five cytokines present within the detection range in our sample set were tested as putative normalization controls.

Algorithms for stability estimation have been thoroughly developed in gene expression studies for the selection of reference genes. We assumed that algorithms which require linear input data could be used to assess the stability of cytokine representation throughout our sample set. Using the pairwise variation-based GeNorm approach [38], we obtained a pair of the best normalization control cytokines, TNF-α and IP-10 (CXCL10). The putative control cytokines were additionally tested for differential representation in the conditioned medium vs. blood plasma. Of the five cytokines, only TNF-α and IP-10 (CXCL10) were not differentially represented in these specimens (Wilcoxon test, p. adj > 0.05). Their mRNA expression was verified in atherosclerotic plaques directly after endarterectomy. The expression of the *TNFA* and *IP-10* (*CXCL10*) genes was detectable in all the specimens tested, confirming their presence in the plaque before ex vivo sub-cultivation. Thus, only TNF-α and IP-10 (CXCL10) were suitable for data normalization in our experimental set.

The blood plasma and the plaque-conditioned medium differed greatly in the secreted cytokine amounts, as evidenced by the number of missed xMAP values: with the same levels of TNF-α and IP-10 (CXCL10) secretion, the plaque explants produced many more other cytokines, which may remain undetected in the plasma due to various reasons (dilution, surface binding, rapid turnover, etc.) Despite this, normalization allowed us to differentiate between the plaque-secreted cytokines and cytokines from other sources. Thus, we confirmed that 22 cytokines out of 41 (IL-1β, IL-1RA, IL-2, IL-4, IL-6, IL-7, IL-8, IL-10, IL-12 (p70), IL-12 (p40), IL-13, IL-15, MCP-1 (CCL2), MIP-1α (CCL3), MIP-1β (CCL4), MCP-3 (CCL7), GRO-α (CXCL1), VEGF, G-CSF, GM-CSF, IFN- α2, and TGF-α) were specifically produced by plaques and poorly represented in the plasma. Meanwhile, the normalized expression of six cytokines (sCD40L, PDGF-AA, PDGF-AB/BB, MDC (CCL22), eotaxin (CCL11), and RANTES (CCL5)) was higher in the plasma compared to the conditioned medium, which implies that there were other sources of these cytokines in blood plasma than the atherosclerotic plaque.

The obtained data provide new insights into the pro- and anti-inflammatory environment within the stable carotid atherosclerotic plaques and highlight the possible interplay of several large molecular signaling pathways that drive inflammation, coagulation, and lipid metabolism [31,39]. For instance, the IL-1β and IL-1RA elevated in our sample set of conditioned media and undetectable in blood plasma represent the signaling molecules of the canonical NLRP3 inflammasome pathway that can be induced by multiple factors, including hypoxia and cholesterol depositions [31,40,41,42]. Their cross-talk is the major regulatory event in atherogenesis. The most notable downstream pro-inflammatory and pro-atherogenic molecules also elevated in our set of plaque-conditioned media include IL-6, IL-8, CCL2, and CXCL1, which are IL-1β-dependent, TNFa-dependent, and PAR-2-dependent [39,43], and are thus overexpressed in response to the initial inflammatory signal and activated coagulation. Apart from CCL2 and CXCL1, the elevated G-CSF and GM-CSF add to the pro-atherogenic cluster of molecules involved in myeloid cell attraction and activation in plaques [44]. However, the effect of these molecules seems to be balanced by the elevated anti-inflammatory cytokines IL-2 and IL-10, which provide endogenous anti-atherogenic signaling [31,39]. All of these molecules along with migration-driving CX3CL1 and CCR5 ligands are of particular interest, as they are currently being evaluated as putative therapeutic targets in atherosclerosis (the roles of the target cytokines, and the present therapeutic approaches and perspectives are reviewed in [31,39,44]). As our study shows, defining such targets at the systemic level is not easy, as some plaque cytokines do not appear in the blood flow. Hence, the ex vivo tissue analysis is more informative. Additionally, the comparison of the secreted cytokines in blood and plaque-conditioned media can provide useful clues to the gradient-forming capacities of chemokines, e.g., the CCR5 ligands.

The expression patterns of the major migration-driving chemokines MCP-1 (CCL2), MIP-1α (CCL3), MIP-1β (CCL4), RANTES (CCL5), and fractalkine (CX3CL1) in patients with atherosclerosis were of particular interest in this study. To closely investigate these, we first confirmed their presence in the immediately endarterectomized plaque at the mRNA level. *MCP1* (*CCL2*), *MIP1A* (*CCL3*), *MIP1B* (*CCL4*), and *FKN* (*CX3CL1*) were expressed in plaque tissue, whereas *RANTES* (*CCL5*) was absent. This was in accordance with the xMAP protein representation and our previous data on mRNA expression in atherosclerotic plaques [29]. The low abundance of the plaque-secreted CCR5 ligands, MIP-1α (CCL3), and MIP-1β (CCL4) in the plasma samples may be explained by their preferential oligomerization and/or binding to the endothelium, which are finely tuned in vivo [45,46]. It is worth noting that another CCR5 ligand, RANTES (CCL5), is elevated in the blood plasma compared to plaque-conditioned media. This implies different sources of these ligands and their different roles in atherogenesis. The interplay between the free and surface-bound ligands of the same receptor may affect cell activation and chemokine gradient migration.

Further analysis of the mRNA from the sorted cells revealed the input of endothelial cells, monocytes/macrophages, and T cells into chemokine synthesis. The *FKN* (*CX3CL1*) gene was not differentially expressed by cell subsets, and *MCP1* (*CCL2*) was predominantly expressed in endothelial cells and monocytes/macrophages of the plaque. Of the CCR5 ligand genes, *MIP1A* (*CCL3*) was predominantly expressed in plaque monocytes/macrophages, whereas *MIP1B* (*CCL4*) was expressed in both monocytes/macrophages and plaque T cells. The expression of *RANTES* (*CCL5*) was restricted to T cells.

Our study has some limitations. First, the model of cytokine secretion in sub-cultivated plaque explants requires verification that the cytokine secretion profile is unbiased by the tissue damage response. The medium was removed after the first day of cultivation, as conducted previously [28], to attenuate the effect of acute tissue damage. We also propose to verify the expression of the cytokines of interest in the plaque directly after endarterectomy using other methods, such as qPCR. Here, we used it for the reference cytokine estimation and for the chemokines that most likely drive immune cell migration to plaques. The xMAP and qPCR results were concordant in both cases. Second, cytokines chosen for normalization may be co-regulated, which may lead to overestimated expression stability and bias in the normalized levels of co-regulated cytokines. Therefore, to estimate a possible bias, co-regulation should be checked prior to the analysis. Importantly, IP-10 (CXCL10) and TNF-α were co-regulated in several other models [47,48]. In our experiment, they fell into the same cluster, indicating their possible co-regulation. Third, owing to the relatively small number of analyzed samples, some differentially represented cytokines may have remained undetected. However, we believe that the sensitivity of the applied methods allowed us to reveal the major differentially expressed cytokines involved in atherogenesis. 

In this work, we developed and tested a method of xMAP data normalization and analysis suitable for the evaluation of systemically represented and plaque-specific secreted cytokine proteins. Our results were verified by qPCR for the selected cytokines involved in immune cell activation and migration (CCL2, CCL3, CCL4, CCL5, CX3CL1), representing potential therapeutic targets in atherosclerosis. The results are consistent with the previously published data [29]. Thus, we indicate a set of plaque-produced cytokines that may drive immune cell activation and migration from blood into atherosclerotic plaques. The cytokines differentially secreted by plaques may potentially serve as markers and/or therapeutic targets in atherosclerosis.

## 4. Materials and Methods

### 4.1. Patients

The study included 34 paired peripheral blood and carotid atherosclerotic plaque specimens collected from patients referred for endarterectomy to the National Medical Research Centre of Surgery named after A.V. Vishnevsky or the Clinical City Hospital named after I.V. Davydovsky. All patients had atherosclerosis of the carotid artery. 

Patient inclusion criteria: indications for carotid endarterectomy [49].

Patient exclusion criteria: acute stroke, transient ischemic attack, or acute myocardial infarction less than 1 month prior to admission; active cancer; autoimmune disease; acute inflammation. The clinical data for the patients are provided in Appendix A (Table A1). According to the criteria, all patients included in the study typically had >60% carotid artery stenosis; only 9 patients out of 34 had previous history of ischemic stroke/transient ischemic attacks (all of them more than 4 months prior to admission). 

Additionally, 9 patients had ischemic heart disease and 6 patients had previous myocardial infarction (all of them more than 1 year prior to admission).

### 4.2. Blood Plasma and Conditioned Culture Medium

Peripheral blood was obtained from patients prior to endarterectomy, collected into S-Monovette^®^ Citrate 3.2% tubes (Sarstedt, Nümbrecht, Germany), and processed less than 20 min after withdrawal. Platelet-poor plasma was obtained via two rounds of centrifugation at 3000× *g* for 15 min (first centrifugation with maximum brake and second centrifugation with brake off). After each centrifugation, the supernatant was transferred to new tubes without disturbing the pellet and carefully pipetted to avoid foaming.

Endarterectomized plaques were collected in RPMI-1640 medium (Gibco, Thermo Fisher Scientific, Waltham, MA, USA) with antibiotic/antimycotic (100 U/mL penicillin, 100 mg/mL streptomycin, 0.25 mg/mL amphotericin B; Gibco, Thermo Fisher Scientific) and processed less than 2 h after the operation. Plaque specimens were washed with fresh RPMI-1640 medium to remove the blood-derived mononuclear cells and cut into circular blocks that were placed for sub-cultivation on a wetted collagen sponge raft (Spongostan Dental by Ethicon, Johnson & Johnson MedTech, Bridgewater, NJ, USA) at the medium–air interface at 37 °C/5% CO_2_, as described earlier [28]. The RPMI-1640 medium (Gibco, Thermo Fisher Scientific) with 15% heat-inactivated fetal bovine serum (HyClone, Cytiva, Grens, Switzerland), antibiotic/antimycotic (100 U/mL penicillin, 100 mg/mL streptomycin, 0.25 mg/mL amphotericin B; Gibco, Thermo Fisher Scientific), MEM non-essential amino acids (Gibco, Thermo Fisher Scientific), sodium pyruvate (Sigma-Aldrich, St. Louis, MI, USA), and GlutaMAX (Gibco, Thermo Fisher Scientific) was used in the experiments. After 24 h of sub-cultivation, the medium was replaced. The plaque explants were then incubated for 72 h more, and the conditioned medium was collected and centrifuged twice at 3000× *g* for 15 min (first centrifugation with maximum brake and second centrifugation with brake off). After each centrifugation, the supernatant was transferred to new tubes without disturbing the pellet and carefully pipetted to avoid foaming.

To ensure that only free soluble cytokines were assessed in the experimental setup, we removed the extracellular vesicles from blood plasma and conditioned medium using ExoQuick (System Biosciences, Palo Alto, CA, USA) for plasma and ExoQuick-TC (System Biosciences, USA) for culture medium according to manufacturer’s instructions. Only the supernatant fraction was taken into analysis. The workflow for xMAP analysis is presented in Figure 1A.

### 4.3. xMAP Cytokine Measurement

We measured the concentrations of 41 cytokines in blood plasma and conditioned explant culture medium using the commercial kit MILLIPLEX MAP Human Cytokine/Chemokine Magnetic Bead Panel (Cat. HCYTMAG-60K-PX41, Merck Millipore, Darmstadt, Germany). The cytokine panel included IL-1α (interleukin-1α), IL-1β (interleukin-1β), IL-1RA (IL-1 receptor antagonist), IL-2, IL-3, IL-4, IL-5, IL-6, IL-7, IL-8, IL-9, IL-10, IL-12 (p40), IL-12 (p70), IL-13, IL-15, IL-17A, MCP-1 (monocyte chemoattractant protein-1 or CCL2), MIP-1α (macrophage inflammatory protein-1α or CCL3), MIP-1β (macrophage inflammatory protein-1β or CCL4), RANTES (or CCL5, regulated on activation, normal T-cell expressed and secreted), MCP-3 (CCL7), eotaxin (CCL11), MDC (macrophage-derived chemokine or CCL22), GRO-α (growth-regulated alpha or CXCL1), IP-10 (interferon-γ-induced protein-10 or CXCL10), fractalkine (CX3CL1), sCD40L (soluble CD40-ligand), EGF (epidermal growth factor), FGF-2 (fibroblast growth factor-2), Flt-3L (Fms-like tyrosine kinase 3 ligand), VEGF (vascular endothelial growth factor), G-CSF (granulocyte colony-stimulating factor), GM-CSF (granulocyte-macrophage colony-stimulating factor), PDGF-AA (platelet-derived growth factor-AA), PDGF-AB/BB (platelet-derived growth factor-AB/BB), TGF-α (transforming growth factor-α), IFN-α2 (interferon-α2), IFN-γ (interferon-γ), TNF-α (tumor necrosis factor-α), and TNF-β (tumor necrosis factor-β).

Prior to analysis, the plasma samples were diluted 1:1 with an assay buffer to reduce the matrix effect. Then, 25 µL of the samples was further diluted with 25 µL of assay buffer. Standards and controls were diluted with appropriate solutions mimicking the properties of the samples (50% serum matrix for plates with plasma, and culture medium for plates with conditioned medium from the plaques). Each sample was then incubated with 15 µL of 41-plex magnetic beads for 18 h at 4 °C. The beads were washed twice with the automatic magnetic washer ELx405 (BioTek, Winooski, VT, USA) and incubated with detection antibodies for 1 h at 25 °C. Antibodies were diluted with wash buffer 1.93 times and added in the amount of 25 µL per well. After incubation, we added 15 µL of Streptavidin-PE solution to the wells and incubated for 30 min at 25 °C. The beads were washed twice, resuspended in the wash buffer, and analyzed using a Luminex 200 instrument (Luminex Corporation, DiaSorin, Saluggia, Italy). We collected 50 beads per region. The fluorescence of blank samples containing culture medium or serum matrix was subtracted from the fluorescence of dilution standards and experimental samples before calculating cytokine concentrations.

During the analysis, we used a 5PL fit for the standard curve. A standard curve was built with 8 standard dilutions (with dilution factor 5 for standard dilutions 2 and 3, and dilution factor 4 for standard dilutions 4–8) made in triplicate. 

### 4.4. xMAP Data Analysis

Cytokine concentrations were assessed in pg/mL according to the standard curves for the cytokines that fell into the detection range. The concentrations of cytokines that fell above or below the detection range were considered initially missing values. Only the cytokines that did not have initially missing values in the sample set were assessed for the lowest variance as putative normalization controls.

The putative normalization controls were tested for the lowest variance in the sample set by calculating their pairwise variation and average representation stability. Pairwise variation (V) was calculated as the standard deviation of the log2-transformed concentration ratios for the pairs of candidate cytokines. The average representation stability in the sample set (M) was calculated for each putative control cytokine as the arithmetic mean of all its pairwise variations. This analysis was performed using the GeNorm algorithm (R 4.0.5, “ctrlGene” package) for gene expression analysis [38], which allows working with linear data and is thus suitable for evaluating absolute cytokine concentrations measured via standard curves. The cytokines were ranked as putative normalization controls according to their average stability (M) values, and pairwise variations were calculated for the cytokines according to their ranking. The normalization factors (NFn) were calculated as the geometric means of control cytokine concentrations, where n is the number of putative control cytokines, ranked from the most stable to the least stable, included in the normalization factor.

For other cytokines, the initially missing concentration values that fell below the lowest standard but could be automatically extrapolated were replaced by the extrapolated values. The imputation technique for the values that fell above the detection range or below the detection range that could not be extrapolated was as follows: In the conditioned medium, the LLOD/2 values (lower limit of detection, the concentration of the maximally diluted standard divided by 2) were used to replace the missing values below the detection range, and the ULOD values (upper limit of detection, concentration of the minimally diluted standard) were used to replace the missing values above the detection range. In the blood plasma, the LLOD values were used to replace the missing values below the detection range, and the ULOD*2 values were used to replace the missing values above the detection range due to the initial dilution of plasma being 1:1. The ULOD and LLOD values are presented in Appendix A (Table A4). 

Cytokine concentrations, normalized by the selected NF, were compared in the plasma and plaque-conditioned media. For cytokines with less than 40% of initially missing and extrapolated values, the normalized levels were compared using the Wilcoxon signed-rank test. For cytokines with more than 40% missing and extrapolated values, the normalized levels were compared using a two-sided sign test. In conditioned medium samples, the missing cytokine concentration values were replaced with the intervals [ULOD; +∞] for values higher than the ULOD, and [0; LLOD] for values lower than the LLOD. In blood plasma samples, the cytokine concentration values were replaced with the intervals [ULOD*2; +∞] for values higher than the ULOD , and [0; LLOD*2] for values lower than the LLOD due to the initial dilution of plasma being 1:1. For cytokines with a measured absolute concentration value, the interval was defined as the value itself. We compared the intervals between groups in pairs and considered the intersecting intervals to be equal values between the groups. We defined the number of pairs where the cytokine concentration was higher/lower between groups and calculated the *p*-value for every cytokine comparison.

### 4.5. Cell Sorting

The paired specimens of atherosclerotic plaques and peripheral blood were obtained from 10 patients undergoing endarterectomy. Peripheral mononuclear cells were isolated from the blood via density centrifugation in CPT tubes (BD, USA). The plaque tissue was finely cut and digested using collagenase IV, as described previously [50]. The obtained cell suspensions were stained with CD3-APC (clone OKT3, BioLegend, San Diego, CA, USA), CD45-PerCP (clone HI30, BioLegend, USA), CD31-PE (clone WM59, BioLegend, USA), CD64-BV510 (clone 10.1, BD Biosciences, Franklin Lakes, NJ, USA), CD68-PE/Cy7 (clone Y1/82A, BD Biosciences, USA), and CD146-AF488 (clone SHM-57, BD Biosciences, USA). The antibodies were titrated prior to experiments, and the spectral compensation matrix was calculated automatically using CompBeads (BD Biosciences, USA). Sorting was performed using a FACSAria SORP instrument (BD Biosciences, USA) with a 70 µm nozzle and corresponding pressure parameters. The endothelial cells were sorted from atherosclerotic plaques based on CD31/CD146 expression, monocytes/macrophages were sorted from atherosclerotic plaques and peripheral blood based on CD64/CD68 expression, and T lymphocytes were sorted from atherosclerotic plaques and peripheral blood based on CD3/CD45 expression. The representative plots are provided in Appendix A, Figure A3. The sorted cells were frozen in 100 µL PBS. For each cell population, at least 4000 events in the corresponding gate were sorted.

### 4.6. RNA Extraction

Total bulk RNA was extracted from 9 finely cut solid plaque tissue specimens and 9 specimens of Ficoll-isolated peripheral blood mononuclear cells obtained from the patients immediately after endarterectomy. The RNeasy mini kit (Qiagen, Hilden, Germany) was used with on-column genomic DNA digestion with Dnase I (Qiagen, Germany), according to the manufacturer’s instructions. For atherosclerotic plaques, a preliminary digestion step with Proteinase K (Qiagen, Germany)—at 55 °C for 10 min—was added before the column transfer, according to the manufacturer’s recommended protocol for RNA isolation from muscle tissue and vessel walls.

Total RNA was extracted from the thawed suspensions of sorted endothelial cells, monocytes/macrophages, and T cells, using a RIBO-prep kit (AmpliSens, Moscow, Russia), according to the manufacturer’s protocol. To remove the genomic DNA, the samples were incubated with 20 units of DNAse I (Sileks, Moscow, Russia) at 37 °C for 10 min and re-extracted using a RIBO-prep kit. 

### 4.7. Real-Time qPCR

To assess the mRNA expression of the *MCP-1* (*CCL2*), *MIP-1A* (*CCL3*), *MIP-1B* (*CCL4*), *RANTES* (*CCL5*), and *FKN* (*CX3CL1*) genes, one-step qPCR was performed using OneTube RT-PCRmix (Evrogen, Moscow, Russia) and the Bio-Rad CFX96 instrument (Bio-Rad, Hercules, CA, USA). The completeness of the genomic DNA removal was tested by performing qPCR without the reverse transcription step. No amplification was detected for all genes tested. The relative RNA amounts of the genes of interest were normalized to the *UBC* reference gene. The primers and probes for *MCP-1* (*CCL2*), *MIP-1A* (*CCL3*), *MIP-1B* (*CCL4*), *RANTES* (*CCL5*), *FKN* (*CX3CL1*), and for the reference gene *UBC* were designed previously [29] and are provided in Appendix A (Table A5). 

To assess the expression of the putative normalization controls, *MDC* (*CCL22*), *IP-10* (*CXCL10*), and *TNFA*, SYBR Green I qPCR was performed. The mRNA was reverse transcribed using the ImProm II kit (Promega, Madison, WI, USA), and qPCR was run using the iTaq Universal SYBR Green Supermix (BioRad, USA) and the Bio-Rad CFX96 instrument (Bio-Rad, USA). The primers for *MDC* (*CCL22*), *IP-10* (*CXCL10*), *TNFA*, and the reference gene *UBC* are provided in Appendix A (Table A6).

### 4.8. Data Analysis

All data were analyzed and plotted using R 4.2.1 and GraphPad Prism 5 (GraphPad Software, Boston, MA, USA) software. The GeNorm function from the “ctrlGene” package was applied. The expression values obtained in the present study were not normally distributed in most cases, according to the Shapiro–Wilk test, and therefore are represented as medians and interquartile ranges [Q1; Q3]. The Wilcoxon signed-rank test was used for comparisons of the normalized cytokine levels in sample sets with less than 40% missing values in the plasma and culture medium and comparisons of chemokine mRNA expression between atherosclerotic plaques and peripheral blood mononuclear cells. A two-sided sign test was used for comparisons of the normalized cytokine levels in sample sets with more than 40% missing values. In order to overcome errors from multiple comparisons of cytokines, we performed a Benjamini–Hochberg FDR correction with the calculation of critical values for each comparison matched with corresponding *p*-values; the adjusted *p*-values were compared with the pre-specified significance level α = 0.05.

## Figures and Tables

**Figure 1 ijms-25-01030-f001:**
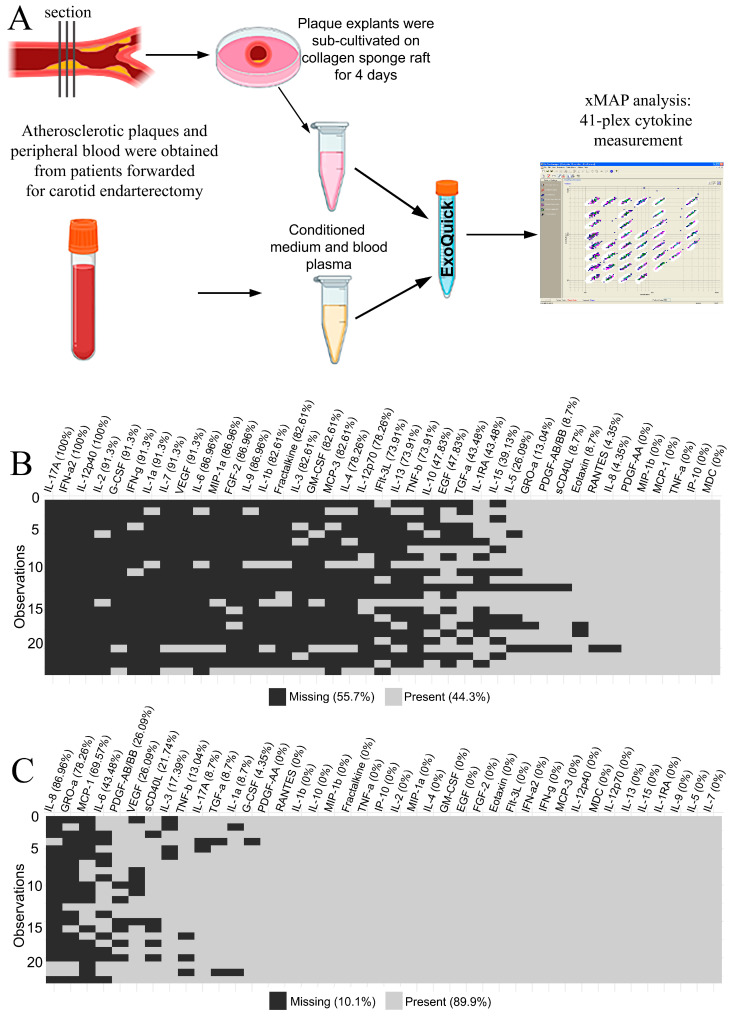
(**A**) The design of the xMAP experiment on blood plasma and plaque-conditioned medium. (**B**) The missing xMAP values in blood plasma specimens. The percentage of missing values is provided in brackets for each cytokine. The total percentage is provided in the legend. (**C**) The missing xMAP values in plaque-conditioned media specimens. The percentage of missing values is provided in brackets for each cytokine. The total percentage is provided in the legend.

**Figure 2 ijms-25-01030-f002:**
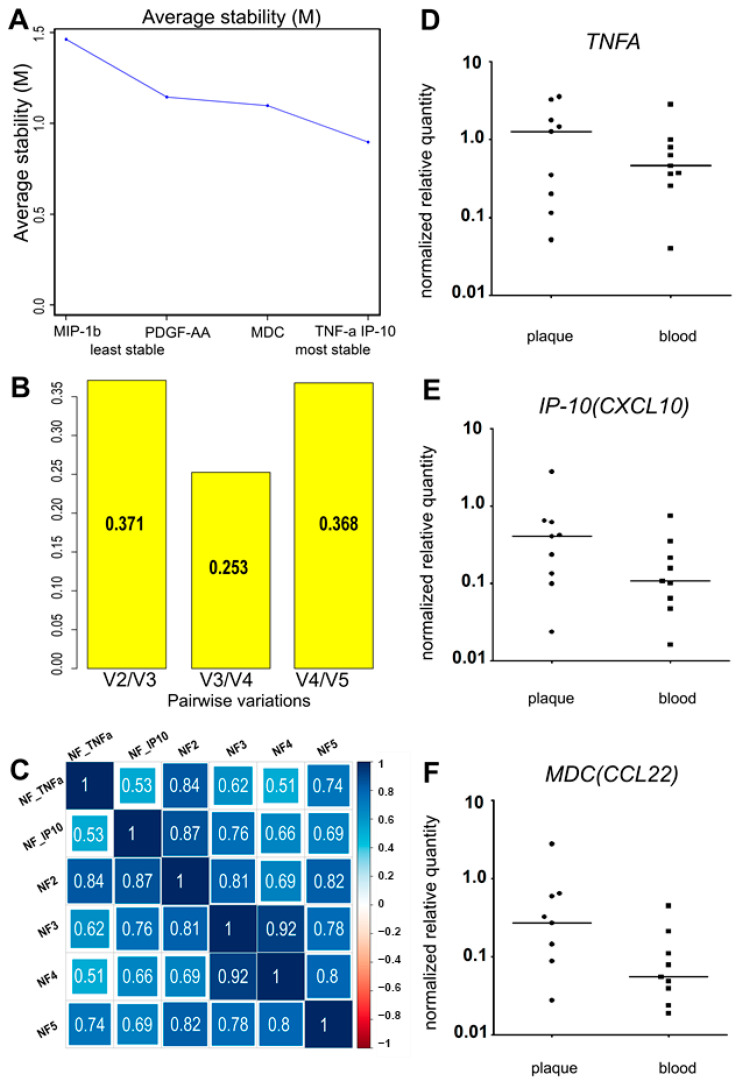
(**A**) The average stability value (M) and the stability ranking of five putative normalization control cytokines. (**B**) The pairwise variations of the ranked putative normalization control cytokines. (**C**) The Spearman correlation coefficients of the calculated normalization factors (with 1; 2; 3; 4; 5 normalization control cytokines). (**D**–**F**). The relative normalized gene expression of three putative reference cytokines in atherosclerotic plaques directly after endarterectomy. The expression levels of the *TNFA* (**D**), *IP-10* (*CXCL10*) (**E**)*,* and *MDC* (*CCL22*) (**F**) genes were analyzed in the bulk mRNA from atherosclerotic plaques (“plaque”, *n* = 9). The data on the bulk mRNA from peripheral blood cells (“blood”, *n* = 9) from the same patients are provided as a positive control. The reference gene *UBC* was used for data normalization. *TNFA* and *IP-10* (*CXCL10*) were expressed in all the specimens analyzed. *MDC* (*CCL22*) was absent in one plaque specimen of nine. No significant differences were found for plaque vs. blood expression levels of the tested cytokines (Wilcoxon test, *p* > 0.05).

**Figure 3 ijms-25-01030-f003:**
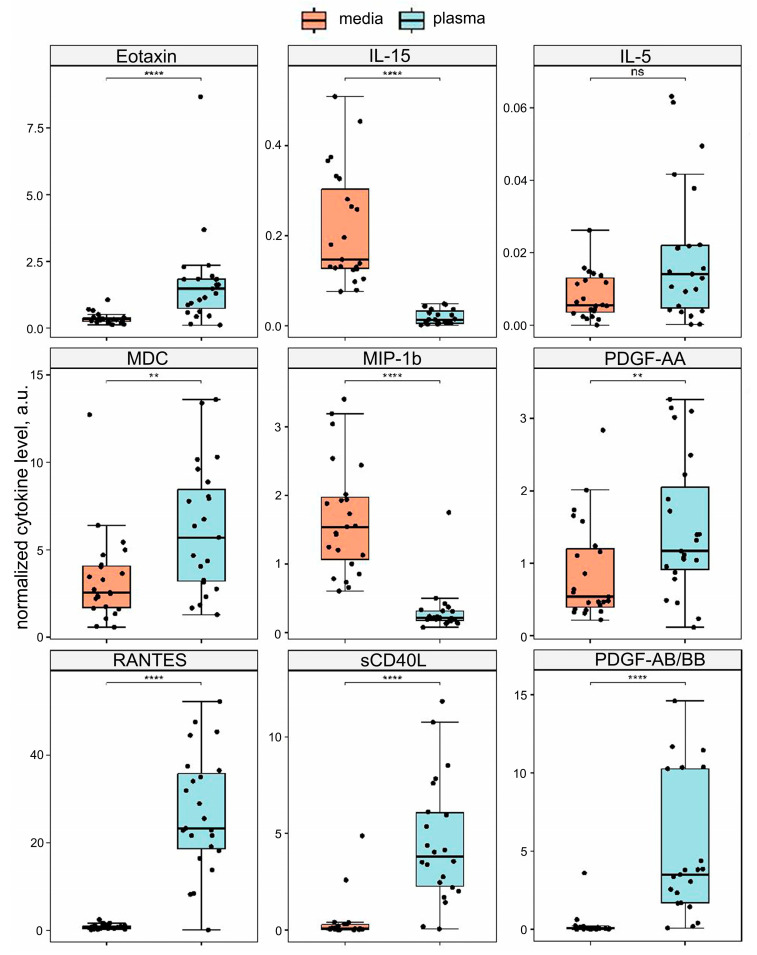
The quantitative analysis of the xMAP data. Cytokine levels were normalized by the geometric mean of the IP-10 (CXCL10) and TNF-α concentrations. The normalized amounts of cytokines are compared in plaque-conditioned medium (medium) vs. blood plasma (plasma). Eotaxin (CCL11), RANTES (CCL5), MDC (CCL22), sCD40L, PDGF-AA, PDGF-AB/BB, IL-15, and MIP-1β (CCL4) are presented differentially (Wilcoxon test, **** — p. adj < 0.0001; ** — p. adj < 0.01; ns —not significant, p. adj > 0.05).

**Figure 4 ijms-25-01030-f004:**
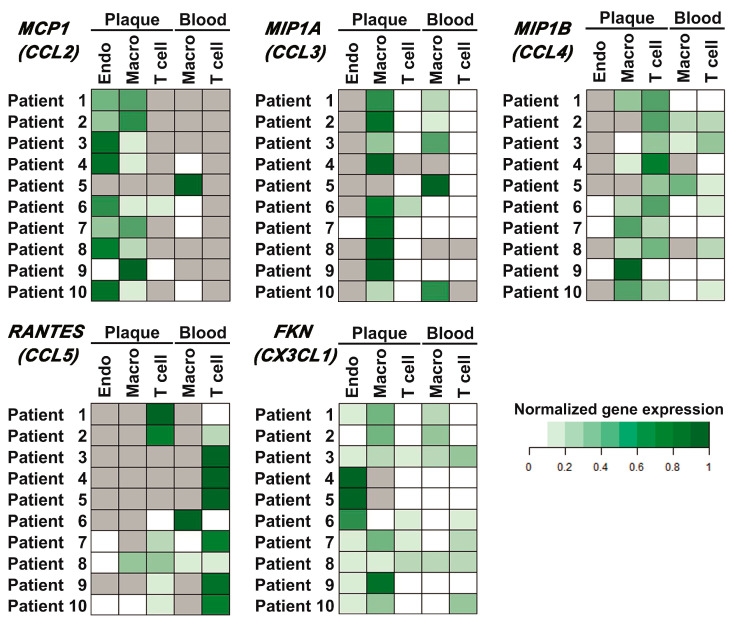
Chemokine gene expression in different cell populations of peripheral blood and atherosclerotic plaques. Endothelial cells (Endo), monocytes/macrophages (Macro), and T cells (T cell) were sorted from the peripheral blood and enzymatically digested atherosclerotic plaques of ten patients, and the mRNA expression levels of the *MCP1* (*CCL2*), *MIP1A* (*CCL3*), *MIP1B* (*CCL4*), *RANTES* (*CCL5*), and *FKN* (*CX3CL1*) chemokine genes were evaluated using real-time qPCR with reverse transcription. The Ct values for chemokine genes were normalized by the ubiquitin C (*UBC*) reference gene. The normalized expression levels (delta Ct) of each chemokine gene for each patient are represented as heatmaps; gray cells indicate cases with undetectable gene expression.

**Table 1 ijms-25-01030-t001:** The normalized cytokine levels compared in blood plasma and conditioned medium via the Wilcoxon signed-rank test (Wilcoxon) or a two-sided sign test (Sign). Of the 41 cytokines, 6 were differentially elevated in the blood plasma, 22 were differentially elevated in the conditioned medium, and 13 were not differentially represented. Cytokines used for normalization are in bold.

Cytokine	Elevated in Plasma	Elevated in Medium	Criterion	Cytokine	Elevated in Plasma	Elevated in Medium	Criterion
EGF	No	No	Sign	IL-4	No	Yes	Sign
Eotaxin	Yes	No	Wilcoxon	IL-5	No	No	Wilcoxon
FGF-2	No	No	Sign	IL-6	No	Yes	Sign
Flt-3L	No	No	Sign	IL-7	No	Yes	Sign
Fractalkine	No	No	Sign	IL-8	No	Yes	Sign
G-CSF	No	Yes	Sign	IL-9	No	No	Sign
GM-CSF	No	Yes	Sign	**IP-10**	**No**	**No**	**Wilcoxon**
GRO-α	No	Yes	Sign	MCP-1	No	Yes	Sign
IFN-α2	No	Yes	Sign	MCP-3	No	Yes	Sign
IFN-γ	No	No	Sign	MDC	Yes	No	Wilcoxon
IL-10	No	Yes	Sign	MIP-1b	No	Yes	Wilcoxon
IL-12p40	No	Yes	Sign	MIP-1α	No	Yes	Sign
IL-12p70	No	Yes	Sign	PDGF-AA	Yes	No	Wilcoxon
IL-13	No	Yes	Sign	PDGF-AB/BB	Yes	No	Wilcoxon
IL-15	No	Yes	Wilcoxon	RANTES	Yes	No	Wilcoxon
IL-17A	No	No	Sign	sCD40L	Yes	No	Wilcoxon
IL-1α	No	No	Sign	TGF-α	No	Yes	Sign
IL-1RA	No	Yes	Sign	TNF-α	No	No	Wilcoxon
IL-1β	No	Yes	Sign	TNF-β	No	No	Sign
IL-2	No	Yes	Sign	VEGF	No	Yes	Sign
IL-3	No	No	Sign				

## Data Availability

The code for the data processing and analysis is provided on GitHub (https://github.com/GeorgeRusakovich/Differential-assessment-of-systemic-and-local-cytokine-levels-in-patients-with-atherosclerosis, accessed on 13 June 2023). The assembled cytokine concentration values are available as .xlsx files.

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
