# Peer review of "Cytokine Profiling of Plasma and Atherosclerotic Plaques in Patients Undergoing Carotid Endarterectomy"

_ijms, 2024, doi:10.3390/ijms25021030_

Round 1

Reviewer 1 Report

Comments and Suggestions for Authors

The cytokines present in blood and in atherosclerotic tissue removed during endarterectomy have been compared. Cytokines are undoubtedly important in atherosclerotic lesion initiation and progression. The authors’ approach is interesting, in that the tissue was cultured and the cytokines measured in the conditioned media and the tissue was then digested and the cells separated by fluorescence-activated cell sorting and the mRNA for the cytokines measured by PCR in the endothelial cells, macrophages/monocytes and T-lymphocytes.

Major points

(1) Twenty three patients were investigated, which is a reasonable number for this type of study. Only 6 were used for the study in which the endothelial cells, macrophages/monocytes and T-lymphocytes were isolated and studied. Given the large variability in the characteristics of atherosclerotic lesions in humans and the results obtained, investigating more lesions would allow more convincing conclusions to be drawn from this part of the research.

(2) The culture medium contained 15% fetal bovine serum. How did the secreted cytokine concentrations compare to the cytokine concentrations already present in the fetal bovine serum?

(3 ) Is it permissible to replace values of cytokines that were above the upper limit of detection by the upper limit of detection values (p. 13, line 447)?

Minor points

It cannot be said that these cytokines was not attributable to atherosclerotic plaques as their plasma levels exceeded those in the conditioned medium (abstract and p. 10, lines 296-298). This might well be true but the concentrations in the conditioned medium would depend on the amount of tissue present and the time of incubation.

p. 12, lines 364, 365 & 372: The concentrations of antimicrobials are very high (10,000 mg of streptomycin/ml is 10 g/ml, which is impossible). The final concentrations should be given.

p. 13, line 3 (and Table A4): What does 1,93 mean? Is it 1.93?

p. 18, Table A4: There are too many significant figures in the data.

Author Response

We thank the Reviewer for the assessment of our work.

We have revised the manuscript accordingly.

Major points

(1)          Twenty three patients were investigated, which is a reasonable number for this type of study. Only 6 were used for the study in which the endothelial cells, macrophages/monocytes and T-lymphocytes were isolated and studied. Given the large variability in the characteristics of atherosclerotic lesions in humans and the results obtained, investigating more lesions would allow more convincing conclusions to be drawn from this part of the research.

We agree with the remark and thus have expanded our qPCR sample set. Of the 11 paired specimens of atherosclerotic plaques and peripheral blood obtained for the qPCR study, 9 were available for bulk RNA analysis; 10 paired specimens of atherosclerotic plaques and peripheral blood were available for cell population sorting. The new Patient data table (Appendix A – Table A1, expanded as suggested by Reviewer 2) provides the clinical data on the patients. 

Please, see the results of our additional qPCRs in the main Figures 2 and 4 and in Appendix B – Figures B2 and B4. Adding new patients did not change the patterns that we observed previously. However, the expanded sample set allowed us to get statistically significant differences in the bulk RNA analysis for the cytokines CCL2, CCL5 and CX3CL1 (Appendix B – Figure B2) and to specify the cell sources for all cytokines of interest: CCL2, CCL3, CCL4, CCL5 and CX3CL1 (Appendix B – new Figure B4). 

(2) The culture medium contained 15% fetal bovine serum. How did the secreted cytokine concentrations compare to the cytokine concentrations already present in the fetal bovine serum?

Thank you for your question. The commercial 41-plex kit for soluble chemokine immunodetection that we are using is human-specific. However, the unspecific detection and the matrix effect caused by FBS that may bias the results are of concern. While we did not directly estimate the cytokine concentrations in FBS we certainly corrected for the bias when measuring. We routinely add culture medium with the same content (including FBS) to the standards and blank wells when using the kit, according to the manufacturer’s protocol. This helps us to compensate for the cytokines present in the culture medium itself and diminish the matrix effect (inhibiting or enhancing effect of proteins and other molecules on the antibody binding to the cytokines that we measure). The fluorescence intensity in blank wells containing culture medium is subtracted from the fluorescence intensities of standards and samples, and this ‘fluorescence minus background’ is further used to calculate secreted cytokine concentrations. We have specified that in Materials and Methods (Lines 489-491; 504-506).

(3 ) Is it permissible to replace values of cytokines that were above the upper limit of detection by the upper limit of detection values (p. 13, line 447)?

Yes, this practice is deemed acceptable when all values exceeding the upper limit of detection are censored data. Employing median imputation or removing values exceeding the detection limit would introduce a larger bias in the actual values compared to the current replacement technique. The major concern of those, who propose to use other imputation techniques is the “downward distribution bias” of the values that exceed the ULOD when they are replaced by ULOD. Although the replacement of such values results in a reduction of the maximum concentrations of cytokines and a decrease in the variability of cytokine concentration values above the detection limit, in our case the question to answer was whether the concentrations of these cytokines are elevated in conditioned media compared to blood plasma. All cytokine samples (G-CSF, GRO-a, IL-6, IL-8, MCP-1) to which this replacement method was applied were from the conditioned media. And all of these cytokines exhibited statistically significant elevation in the media (comparison of normalized concentrations by two-sided Sign test, please refer to Table 1 for the results on these cytokines). Consequently, while the replacement method reduced the maximum values in the samples, even if the true values were known, the comparison between groups would yield similar results.

Thus, in our specific scenario, excluding values or alternative imputation methods that would require models to predict values exceeding the ULOD are not viable. However, even with the current replacement technique that potentially reduces the values in our sample sets we obtained statistically significant elevation of cytokines in them and were able to answer our question.

Minor points

It cannot be said that these cytokines was not attributable to atherosclerotic plaques as their plasma levels exceeded those in the conditioned medium (abstract and p. 10, lines 296-298). This might well be true but the concentrations in the conditioned medium would depend on the amount of tissue present and the time of incubation.

Thank you and we agree with this. These 6 cytokines’ levels (namely, sCD40L, PDGF-AA, PDGF-AB/BB, MDC (CCL22), eotaxin (CCL11), and RANTES (CCL5)) may be partly attributable to atherosclerotic plaques but not only to them, therefore we amended the text in the Abstract (Line 23) and Discussion (Line 339-340). We used normalization by two reference cytokines (TNFa and IP-10) to compensate for the effect of the amount of tissue and the time of incubation. So basically we conclude that similar amounts of the reference cytokines correspond to larger amounts of these 6 cytokines in blood plasma and smaller amounts of these cytokines in plaque-conditioned medium. Based on this, there have to be other sources of these cytokines in plasma then just the plaque alone.

  1. 12, lines 364, 365 & 372: The concentrations of antimicrobials are very high (10,000 mg of streptomycin/ml is 10 g/ml, which is impossible). The final concentrations should be given.

Thank you for your observation. There was a mistake in this line, the final concentrations are 100 units/mL of penicillin, 100 µg/mL of streptomycin, and 0.25 µg/mL amphotericin B. We have corrected this information in the article (Lines 446-455).

  1. 13, line 3 (and Table A4): What does 1,93 mean? Is it 1.93?

Yes. This is 1.93. We corrected the decimal in the Materials and Methods (Line 494) and all throughout Table A4.

  1. 18, Table A4: There are too many significant figures in the data.

The LLOD and ULOD values are technical data from the runs and are exported from the software rounded to two decimal places. We kept this technical data in Table A4  in this format for accuracy.

Please, find the amended manuscript attached.

Reviewer 2 Report

Comments and Suggestions for Authors

In this fascinating study Potashnikova er al. demonstrated the expression patterns of some important chemokines in endarterectomized plaques via qPCR  and identify the cells responsible for the production of the aforementioned chemokines within the. This experimental study is well conducted but some issues should be solved to improve the presentation and comprehensibility of the study. Below are some comments and suggestions:

-       the order used for the paragraphs of the text can cause confusion for the reader. I recommend using the following order: introduction, the aim of the study, materials and methods, results, and discussion. I also recommend inserting a specific paragraph for the conclusions preceded by a paragraph with the limitations of the study.

-       You only included “indications for carotid endarterectomy” as inclusion criteria. It would be interesting to have the details of surgical indications for all patients. Were all symptomatic plaques or asymptomatic plaques determining severe stenosis? Since you added acute stroke as an exclusion criterion, it is reasonable to think that you did not recruit patients with symptomatic plaques. Please clarify inclusions and exclusions criteria with details about clinical carotid plaque characteristics.

-       The discussion is specific about the experiment but vague about its possible scientific impact. What is your interpretation of the different patterns of chemokines encountered in plasma and endarterectomy plaque? What possible impact could have on future plaque studies? Since the role of the chemokines studied, I suggest discussing briefly the possible interaction between coagulation and inflammation in carotid plaques and arterial wall.  (Miceli G, Basso MG, Rizzo G, Pintus C, Tuttolomondo A. The Role of the Coagulation System in Peripheral Arterial Disease: Interactions with the Arterial Wall and Its Vascular Microenvironment and Implications for Rational Therapies. Int J Mol Sci. 2022 Nov 29;23(23):14914. doi: 10.3390/ijms232314914. PMID: 36499242; PMCID: PMC9739112.)

Author Response

In this fascinating study Potashnikova er al. demonstrated the expression patterns of some important chemokines in endarterectomized plaques via qPCR  and identify the cells responsible for the production of the aforementioned chemokines within the. This experimental study is well conducted but some issues should be solved to improve the presentation and comprehensibility of the study. Below are some comments and suggestions:

-       the order used for the paragraphs of the text can cause confusion for the reader. I recommend using the following order: introduction, the aim of the study, materials and methods, results, and discussion. I also recommend inserting a specific paragraph for the conclusions preceded by a paragraph with the limitations of the study.

We thank the Reviewer for the high estimation of our work. Unfortunately, the paragraph order is determined by the journal and the template for the article it provides. The format of IJMS and their template we are using puts the Materials and Methods in the end of the manuscript, we cannot replace this. However, to make our points more clear and easy to read, we specified the aim of the study in the Introduction (Lines 72-75) and expanded the last paragraph of Discussion to conclude the manuscript.

-       You only included “indications for carotid endarterectomy” as inclusion criteria. It would be interesting to have the details of surgical indications for all patients. Were all symptomatic plaques or asymptomatic plaques determining severe stenosis? Since you added acute stroke as an exclusion criterion, it is reasonable to think that you did not recruit patients with symptomatic plaques. Please clarify inclusions and exclusions criteria with details about clinical carotid plaque characteristics.

Thank you for your question. The information about the patients was presented in the renewed Table A1 (Appendix A) and Lines 427-432. Please note that the sample set was expanded (as demanded by Reviewer 1) and now contains 34 specimens. Specimens ##1-23 were used for Luminex analysis, specimens ##24-34 were used for qPCR. The clinical characteristics for the specimens used in different experiments were matched, as possible. Most of the patients included in the study had no previous history of stroke or ischemic attacks, and all plaques invariably had high % of stenosis, which was the main indication for operation, so the plaques included in the experiment may be called asymptomatic with severe stenosis. Additionally, we provided the information on the signs of cap rupture (where it was available from CT description or surgical protocols) to assess the thrombogenic potential of the plaques. As only 8 plaques showed signs of cap rupture (only 2 of them – at preliminary computed tomography) the plaques included in the study may be described as stable.

-       The discussion is specific about the experiment but vague about its possible scientific impact. What is your interpretation of the different patterns of chemokines encountered in plasma and endarterectomy plaque? What possible impact could have on future plaque studies? Since the role of the chemokines studied, I suggest discussing briefly the possible interaction between coagulation and inflammation in carotid plaques and arterial wall.  (Miceli G, Basso MG, Rizzo G, Pintus C, Tuttolomondo A. The Role of the Coagulation System in Peripheral Arterial Disease: Interactions with the Arterial Wall and Its Vascular Microenvironment and Implications for Rational Therapies. Int J Mol Sci. 2022 Nov 29;23(23):14914. doi: 10.3390/ijms232314914. PMID: 36499242; PMCID: PMC9739112.)

We thank the Reviewer for the suggestion. A paragraph (Lines 341-364) was added to describe the most interesting findings and their impact on the future studies. The main point is that the capacity of the plaque-produced cytokines to enter circulation has been insufficiently addressed so far. And the systemic blood plasma estimations, that are fairly popular, fail to address the local signaling pathways in full. All the major atherogenic cytokines (except, probably, CCL2) that represent the most interesting potential therapeutic targets are poorly represented in plasma and to assess their validity - ex vivo detection in plaque tissue must be employed. The methods of such detection may include ex vivo secretion assays by tissue explants, as we show with some additional testing. As we specifically focused on CX3CL1 and CCR5 ligands in this study, we were very interested in the differential representation of CCL5 and CCL3/CCL4, with CCL5 being out in free circulation and very moderately produced by the plaque and CCL3 being inside the plaque (specifically produced by macrophages) as well as CCL4 (produced by macrophages and T cells alike). As we mention in Line 377 – this implies different sources of these ligands to one receptor and their differential roles in atherogenesis.

Please, find the amended manuscript attached.

Round 2

Reviewer 1 Report

Comments and Suggestions for Authors

The authors have carried out additional experiments and have improved the manuscript. I have no further comments.

Reviewer 2 Report

Comments and Suggestions for Authors

Thank you for your response. I do not have any other comments.